# Does Hospitalist Care Enhance Palliative Care and Reduce Aggressive Treatments for Terminally Ill Patients? A Propensity Score-Matched Study

**DOI:** 10.3390/cancers15153976

**Published:** 2023-08-04

**Authors:** Nin-Chieh Hsu, Chun-Che Huang, Chia-Hao Hsu, Tzung-Dau Wang, Wang-Huei Sheng

**Affiliations:** 1Division of Hospital Medicine, Department of Internal Medicine, College of Medicine, National Taiwan University, Taipei 10051, Taiwan; chesthsu@gmail.com (N.-C.H.); tdw0911@gmail.com (T.-D.W.); 2Division of Hospital Medicine, Department of Internal Medicine, Taipei City Hospital Zhongxing Branch, Taipei 103212, Taiwan; 3Department of Healthcare Administration, College of Medicine, I-Shou University, Kaohsiung 84001, Taiwan; huangmtc@gmail.com; 4Department of Orthopedics, Kaohsiung Medical University Hospital, Kaohsiung Medical University, Kaohsiung 80708, Taiwan; 5Division of Cardiology, Department of Internal Medicine, National Taiwan University Hospital, Taipei 100229, Taiwan; 6College of Medicine, National Taiwan University, Taipei 10051, Taiwan; whsheng@ntu.edu.tw

**Keywords:** terminally ill, hospitalist, palliative care, hemodialysis, life-sustaining treatment

## Abstract

**Simple Summary:**

Patients with severe diseases at the end-of-life stage are mostly hospitalized and may receive care from a hospitalist in the current healthcare system. In this study, we found that hospitalist care may enhance palliative care and reduce unnecessary aggressive life-sustaining treatment, such as mechanical ventilation, tracheostomy, surgery, and intensive care unit transfer before death. Although the benefit of hospitalist care at the end-of-life stage requires further validation, it potentially improves care for terminally ill patients in the hospital, especially when palliative care services are scarce.

**Abstract:**

Background: Information on the use of palliative care and aggressive treatments for terminally ill patients who receive care from hospitalists is limited. Methods: This three-year, retrospective, case-control study was conducted at an academic medical center in Taiwan. Among 7037 patients who died in the hospital, 41.7% had a primary diagnosis of cancer. A total of 815 deceased patients who received hospitalist care before death were compared with 3260 patients who received non-hospitalist care after matching for age, gender, catastrophic illness, and Charlson comorbidity score. Regression models with generalized estimating equations were performed. Results: Patients who received hospitalist care before death, compared to those who did not, had a higher probability of palliative care consultation (odds ratio (OR) = 3.41, 95% confidence interval (CI): 2.63–4.41), and a lower probability to undergo invasive mechanical ventilation (OR = 0.13, 95% CI: 0.10–0.17), tracheostomy (OR = 0.14, 95% CI: 0.06–0.31), hemodialysis (OR = 0.70, 95% CI: 0.55–0.89), surgery (OR = 0.25, 95% CI: 0.19–0.31), and intensive care unit admission (OR = 0.11, 95% CI: 0.08–0.14). Hospitalist care was associated with reductions in length of stay (coefficient (B) = −0.54, 95% CI: −0.62–−0.46) and daily medical costs. Conclusions: Hospitalist care is associated with an improved palliative consultation rate and reduced life-sustaining treatments before death.

## 1. Introduction

For general internal medicine hospitalization, hospitalist care models have recently grown in developed and developing countries, including Taiwan [1], while their pros and cons are still debated [1,2]. To care for a majority of patients with complex and multiple comorbidities from emergency departments, Taiwan established a hospitalist care model for acute general medicine admissions in late 2009 [3]. The efficiency and quality of patient care with hospitalist services have been investigated extensively, suggesting that a shorter length of stay (LOS) and reduced hospital costs and malpractice premiums [3,4] were associated with admitted patients who received care from hospitalist teams. However, the outcomes of complications, in-hospital mortality, readmission, and patient satisfaction remain inconclusive [4].

The utilization of palliative care services is essential for patients with terminal illnesses, including cancers [5]. Terminally ill patients may experience altered mental status and are unable to make their own decisions [6]. With advances in medical and technological equipment, terminally ill patients who experience life-threatening conditions at the end-of-life (EOL) stage may be subjected to aggressive and unnecessary medical intervention [7]. Patients with terminal illnesses may be more likely than patients with stable conditions to receive excessive medical and surgical treatment during their last hospitalization. A previous study from Canada reported that 35% of EOL patients who requested not to be resuscitated were ordered to receive treatments, including cardiopulmonary resuscitation (CPR) and other life-supporting measures [8]. In addition, family member participation was associated with a higher risk of potential overtreatment for terminally ill patients at the EOL [8]. Berge et al. indicated that unrealistic family expectations for extremely ill patients admitted to the intensive care unit (ICU) correlated with increased aggressive or expensive resource utilization without a one-year survival benefit [9]. To deal with these well-known barriers and challenges, it is important to identify the facilitators, especially at the provider level [10]. Decisions regarding life-sustaining treatment are complex, involving the patient’s age, educational level, and stage of disease [11]. From preference discussion to advance directive, an in-charge physician with a multidisciplinary team is necessary.

As hospitalists expand their roles, palliative care intervention, advance directive documentation, and communication between medical personnel, patients, and patients’ families have been reported [12]. However, to our knowledge, the utilization of palliative care and hospital resources by hospitalists for patients with terminal illnesses has not been well studied. Therefore, this study aims to assess whether hospitalist care is associated with improved palliative care intervention and reduced aggressive life-sustaining procedures and hospital resources during the EOL among terminally ill patients.

## 2. Materials and Methods

### 2.1. Data Source

This retrospective case-control study used data from electronic medical records of patients who were admitted to a tertiary care hospital, National Taiwan University Hospital (NTUH), in northern Taiwan, between 1 June 2012 and 31 December 2015. A well-designed hospitalist program was developed in 2009, where details on staffing, service model, and performances are published in the literature [3]. Hospitalists at NTUH work exclusively in the hospitalist wards and do not take care of patients outside. The decision to admit patients to hospitalist wards is made by emergency physicians rather than being selected by hospitalists [3]. This mitigates the selection bias of clinical severity and allows a fair comparison of outcomes and performances with other departments. Patients admitted to the hospitalist ward encompassed a wide range of general medical conditions, including pneumonia, urinary tract infection, chronic obstructive pulmonary disease, gastrointestinal bleeding, cholecystitis, and pancreatitis. Hospitalists served as the attending physicians for patients in hospitalist wards throughout the whole hospitalization course. There is no co-management program with surgeons in this model. The design of the hospitalist program remained unchanged during the study period.

This study’s protocol was reviewed and permitted by the Research Ethics Committee of the NTUH (approval no. 201502011RINC), who waived the requirement to obtain informed consent from patients. In addition, the retrospective study design complied with the legal regulations for data privacy protection. All personal identifiable information was encrypted and de-identified to protect patient privacy and confidentiality before being released for research.

### 2.2. Study Population

We identified 7674 patients who died in hospital or were discharged to die at home after being hospitalized. Patients whose LOS was less than 48 h (*n* = 637) were excluded because the short stay would have limited the timeliness of any intervention. Of the remaining 7037 patients, 815 patients who received care from hospitalists were defined as the case group. A control group was created from 6222 patients who did not receive hospitalist care. To reduce the differences in baseline characteristics between the two groups, propensity score matching was performed at a 1:4 ratio, and 815 cases and 3260 controls were included in this analysis. Figure 1 describes the flow diagram of patient selection.

### 2.3. Outcomes and Measurements

The primary outcomes were (1) palliative care consultation before death and (2) the use of life-sustaining treatment procedures during hospitalization, including invasive mechanical ventilation (IMV), non-invasive ventilation (NIV), tracheostomy, hemodialysis, and surgical intervention, as well as transferring to the ICU.

The hospital-based palliative care consultation that has been actively promoted in Taiwan since 2005 was developed to provide terminally ill patients with comprehensive EOL care in acute care wards by a qualified interdisciplinary team of specialists [13]. Patients and families can choose freely to undergo palliative care consultation and make EOL decisions. Palliative care consultation was conducted by attending physicians using formal discussions and shared decision-making with terminally ill patients and their families regarding EOL care goals, hospice care, and preferred place of death. The IMV, NIV, tracheostomy, hemodialysis, surgery, and ICU transfer that patients with terminal illness would probably undergo during the EOL hospitalization were measured as proxies for the use of life-sustaining procedures.

The secondary outcomes were the LOS and total medical costs during the EOL hospitalization, which were used as proxies to allow comparisons of the utilization of hospital resources between groups. LOS was evaluated as the total number of days per hospital stay, from admission to death or discharge. The total medical costs during the hospitalization before death for each patient were calculated based on the care received between initial admission and death or discharge to die at home. In addition, the average daily medical cost per patient was calculated by dividing the total medical costs by the LOS.

The independent variable was whether the terminally ill patient had received hospitalist care during the EOL hospitalization. In late 2009, NTUH, in northern Taiwan, established a pioneer hospitalist program for acute general medicine admissions, which introduced hospitalists to the traditional inpatient care model that provided equal quality of care for the duration of hospitalization [3,14].

Patient’s age and gender, catastrophic illness certificate, principal diagnoses, comorbidities, and disposition status were included as covariates. Patients who were diagnosed with cancers, chronic psychiatric diseases, hemodialysis, or congenital disorders are eligible for catastrophic illness certificates after a review by the National Health Insurance Administration (NHIA) in Taiwan. Insured patients with a catastrophic illness certificate are entitled to a waiver for medical co-payments. In addition, based on the International Classification of Diseases, Clinical Modification, Ninth Revision (ICD-9-CM) coding system, the principal diagnoses were categorized as follows: infectious and parasitic diseases (ICD-9-CM codes 001–139); neoplasms (140–239); diseases of the circulatory system (390–459); diseases of the respiratory system (460–519); diseases of the digestive system (520–579); diseases of the genitourinary system (580–629); injury and poisoning (800–999), and other diagnoses. The severity of comorbidities for each patient was applied using the Deyo-modified Charlson comorbidity index (CCI) [15]. The disposition status of terminally ill patients was categorized into death in hospital or discharged to die at home. Given the Chinese cultural preference for dying at home, patients are often offered oxygen or respiratory support to enable them to pass away in their homes, rather than in hospitals, during their final moments of life [16]. As the time spent at home was typically brief, these patients were often considered to have passed away in the hospital, with their date of discharge from the most recent hospitalization being recorded as their date of death.

### 2.4. Statistical Analysis

All statistical analyses were performed using SAS version 9.4 (SAS Institute, Cary, NC, USA). The distributions of demographic data, medical conditions, and utilization of life-sustaining treatments during the final hospitalization before death between the case and control groups were compared using chi-square or Fisher’s exact tests for categorical variables and student’s *t*-test for continuous variables, as appropriate. In addition, to reduce the treatment selection bias and potential confounding effect between the two groups, matching at a 1:4 ratio was performed using a propensity score based on age, gender, catastrophic illness, and CCI score.

Odds ratios (ORs) and regression coefficients (B) with 95% confidence intervals (CIs) for the associations between hospitalist program status and utilization of life-sustaining procedures and hospitalization resources during the EOL were estimated using univariable and multivariable regression analyses. To account for the clustering of patients hospitalized on the same date, generalized estimating equation (GEE) models with exchangeable correlation structures were conducted. A *p*-value less than 0.05 was set as the level of statistical significance.

## 3. Results

### 3.1. Demographics and Outcomes

#### 3.1.1. Characteristics of Cohorts

A total of 7037 deceased patients who died of terminal illnesses were identified during the study period, of which 815 patients (11.6%) received a hospitalist program during the EOL hospitalization. The baseline characteristics of terminally ill hospitalized patients before and after propensity score matching are presented in Table 1. Overall, 41.7% had a primary diagnosis of cancer. A comparison of characteristics between the unmatched patients who did not receive a hospitalist program showed significant differences in age, catastrophic illness, and CCI score before propensity score matching. In addition, terminally ill patients who received hospitalist care experienced a significantly higher proportion of infectious and parasitic, respiratory system, digestive system, and genitourinary system diseases than those who did not. This indicates that further matching is necessary.

#### 3.1.2. Propensity Score Matching

A propensity score was generated using logistic regression based on significant variables, including age, gender, catastrophic illness, and CCI score. After propensity score matching, patient characteristics were well matched in both groups with respect to age, gender, catastrophic illness, and CCI score. However, patients who received hospitalist care still had a higher proportion of infectious and parasitic (9.7% vs. 7.4%, *p* = 0.035), respiratory system (29.9% vs. 18.7%, *p* < 0.001), digestive system (9.8% vs. 6%, *p* < 0.001), and genitourinary system diseases (4.7% vs. 2.7%, *p* = 0.004) compared with those who received non-hospitalist care, respectively. Meanwhile, patients who received hospitalist care had a significantly lower proportion of neoplasms (29.8% vs. 36.2%, *p* < 0.001), circulatory system diseases (6.1% vs. 14.9%, *p* < 0.001), and occurrences of injury and poisoning (1.7% vs. 3.2%, *p* = 0.02), respectively (Table 2).

In terms of the process of care during the last hospitalization, patients who received hospitalist care had a higher proportion of palliative care consultation before death (14.9% vs. 5%, *p* < 0.001) than those who did not, respectively (Table 3). However, the crude proportions of undergoing IMV (9.4% vs. 38.9%, *p* < 0.001), NIV (8% vs. 16.1%, *p* < 0.001), tracheostomy (0.7% vs. 4.6%, *p* < 0.001), hemodialysis (10.9% vs. 14.5%, *p* = 0.009), and surgery (10.8% vs. 33.3%, *p* < 0.001) were lower for patients who receive hospitalist care, respectively.

Furthermore, the proportion of transfers to the ICU (10.8% vs. 45.8%, *p* < 0.001) during hospitalization was significantly lower among patients who received hospitalist care compared to those who did not, respectively. There was a shorter LOS (median of 12 days vs. 19 days, *p* < 0.001) and lower total (median of (Taiwan dollar) TWD 65,590 vs. TWD 192,745.5, *p* < 0.001) and daily medical costs (median of TWD 6129.6 vs. TWD 9739.5, *p* < 0.001) in patients who received hospitalist care than those who did not, respectively (Table 3).

#### 3.1.3. Regression Analyses

Table 4 depicts the univariable and multivariable regression analyses for primary and secondary outcomes associated with hospitalist care as compared with non-hospitalist care. In the univariable regression model, all the primary and secondary outcome variables were statistically significant. Palliative care consultation was positively associated with hospitalist care (OR = 3.29, 95% CI: 2.56–4.24). However, life-sustaining procedures (invasive and non-invasive mechanical ventilation, tracheostomy, hemodialysis, surgery) and ICU admission showed a negative association with hospitalist care.

Multivariable analyses were performed with adjustments for age, gender, catastrophic illness, primary diagnoses, CCI score, and disposition status (Table 3). Compared to those who received non-hospitalist care, terminally ill patients who received hospitalist care were more likely to have palliative care consultation (OR = 3.41, 95% CI: 2.63–4.41). However, a lower probability of undergoing IMV (OR = 0.13, 95% CI: 0.10–0.17), NIV (OR = 0.51, 95% CI: 0.39–0.66), tracheostomy (OR = 0.14, 95% CI: 0.06–0.31), hemodialysis (OR = 0.70, 95% CI: 0.55–0.89), and surgery (OR = 0.25, 95% CI: 0.19–0.31), as well as being transferred to the ICU (OR = 0.11, 95% CI: 0.08–0.14), was shown in the hospitalist care group as compared to the non-hospitalist care group. In addition, the associated reductions in LOS (B = −0.54, 95% CI: −0.62–−0.46) and total (B = −0.96, 95% CI: −1.06–−0.87) and daily medical costs (B = −0.59, 95% CI: −0.65–−0.54) were statistically significant in patients who received hospitalist care after adjustments (Table 4).

## 4. Discussion

This study is the first investigation to examine the effects of hospitalist care on palliative care consultation and utilization of life-sustaining treatments and hospital resources during EOL hospitalization. Our results indicate that terminally ill hospitalized patients who received hospitalist care were more likely than those who received non-hospitalist care to have palliative care consultation before death, after adjusting for demographic characteristics, co-morbidities, and disease severity. These findings are consistent with that of a previous study that showed a positive correlation between hospitalist care and the provision of palliative care consultation among cancer and non-cancer hospitalizations [17]. Hospitalist care may have acquired the expertise and skills of palliative care and worked closely with a specialist palliative care team.

Terminally ill conditions are complex and heterogeneous, which makes a prospective or randomized design difficult. In the present quasi-experimental study using propensity score matching, patients who received hospitalist care had a significantly lower probability of undergoing invasive mechanical ventilation, non-invasive ventilation, tracheostomy, hemodialysis, and surgery, as well as of being transferred to the ICU before death. Care modalities during the EOL stage may be intensive, aggressive, and high-tech, but they have unproven benefits. Some of these treatments may be negatively correlated with patient preferences and introduce physical and psychological burdens [7]. Some interventions have shown an impact on reducing futile resuscitation before death. A previous study reported that patients who have advance directives or do not resuscitate orders were less likely to receive cardiopulmonary resuscitation at EOL [11]. However, the effect of an intervention on a more comprehensive list of life-sustaining treatments is scarcely reported.

Furthermore, for terminally ill patients who received hospitalist care, the LOS and total/daily medical costs decreased significantly in our study. The results were consistent with previous studies that almost universally confirmed that hospitalist team care was associated with shorter LOS and reduced total hospital costs for hospitalized patients [18]. The reduction in daily medical costs confirmed that cost reduction was not solely due to shorter LOS. However, there has been great variability in the practice patterns of hospitalists worldwide [19,20]. In addition, the learning curve, experience, and available support of hospitalist teams may influence the process and utilization of aggressive intensive care [21]. Future research is required to revalidate our findings.

Our findings may imply that a hospitalist-led multidisciplinary team approach for terminally ill patients is the key to enhancing palliative care and reducing the utilization of life-sustaining procedures and hospitalization resources during the EOL stage. Timely palliative care for terminally ill patients is imperative. The quality of end-of-life care should not solely rely on physician attitudes but should also encompass the involvement of patients and their families in treatment decision-making. This requires a well-designed system to trigger a referral when patients meet the criteria [22]. In hospitals with a scarce palliative care resource, a hospitalist program may maximize the opportunity for palliative care for patients at the EOL stage. A routine and early palliative care initiation to deal with the burden of physical and psychological symptoms of patients with malignant disease has become the standard of care [23]. Furthermore, the importance of palliative care intervention for patients with end-stage organ dysfunction, such as kidney injury [24,25], advanced heart failure [26], and chronic obstructive pulmonary disease [27], has been increasingly addressed. Discussions of EOL care preferences and modalities of life-sustaining treatments are challenging [28]. As generalists, hospitalists may be the most suitable candidates to care for these non-malignant EOL scenarios.

In our study, a significant reduction in ICU transfer before death was noticed under hospitalist care. Patients who suffer from acute complications or require continuous monitoring may benefit from intermediate care by the multidisciplinary management model [29]. However, ICU admission is not beneficial for most patients at EOL stages. Prolonged mechanical ventilation is common among frail patients who undergo endotracheal intubation because of abnormal respiratory drive or ventilator-induced diaphragm dysfunction [30]. Although palliative care intervention and discussions of life-sustaining treatments are often performed in the ICU, the studies were mostly reported in North America in a recent systematic review [31]. In addition, it has proven difficult to affect the families’ and surrogates’ burden of psychological symptoms in the ICU setting [32]. Terminally ill patients who received hospitalist care may carry a higher probability of early palliative care intervention, reducing aggressive intensive care, and may potentially decrease symptom and psychological burdens.

This study strengthens the evidence for a connection between hospitalist care and the use of palliative care and life-sustaining procedures during the EOL stage. Typical hospitalist care services for inpatients with comorbid conditions were organized and conducted by multidisciplinary teamwork. This can facilitate better co-management with palliative care specialists. Moreover, this study uses the propensity score matching to account for baseline differences between terminally ill patients with and without receiving hospitalist care, thereby reducing the effects of inherent bias and treatment variation in an observational study.

In East Asia, palliative care is progressing, and education in the field is extended from physicians to nurses and paramedical professionals [33,34]. Physicians such as intensive care specialists who take care of critically ill patients should also be engaged [35]. The traditional beliefs and perceptions of cancer and death between the East and the West differ significantly, which requires a systemic approach by the care delivery system [36]. The integration of hospitalist care within the existing healthcare system has the potential to enhance the provision of palliative care to patients in a synergistic manner [37,38].

Several limitations of this study should be mentioned. First, there were different sample sizes between patients with and without receiving hospitalist care. Nevertheless, this case-control study represented results with no difference in patient’s age, gender, catastrophic illness, and CCI score between the two groups using matching. It is probably more convincible than adjustments using regression models. Second, unmeasured factors that influence the observed association between the two groups may exist in a retrospective design. However, the propensity score matching for large sample size in our study provides more robust results. The univariate regression analyses showed a uniformly significant reduction in all life-sustaining procedures. Although prospective randomized design for terminally ill patients is challenging, further trials are mandatory to revalidate our findings. Third, the generalizability of the results from a single center may be limited to other medical centers and community hospitals. Hospitalist programs in other clinical settings may have differences in services and practice patterns, but they usually learn from each other using quality improvement initiatives. Fourth, exposure to hospitalist care, palliative care consultation, and outcomes regarding life-sustaining procedures are all dichotomous factors that make sensitivity analyses impossible. Fifth, details regarding the sorts and stages of chronic organ failure and malignancy are unavailable in our study. We therefore cannot overstate the findings of any disease entities.

## 5. Conclusions

Terminally ill patients receiving hospitalist care are positively correlated with the use of palliative care consultation and had a lower likelihood of undergoing IMV, NIV, tracheostomy, hemodialysis, and surgery, as well as of being transferred to the ICU during the EOL stage. Our findings suggest that the hospitalist program could promote palliative care consultation service and reduce the utilization of aggressive life-sustaining procedures and inefficient hospitalization resources before patients die from a terminal illness. Additional research is required to clarify the essential elements of hospitalist care that contribute to effective palliative care for patients who are terminally ill and hospitalized.

## Figures and Tables

**Figure 1 cancers-15-03976-f001:**
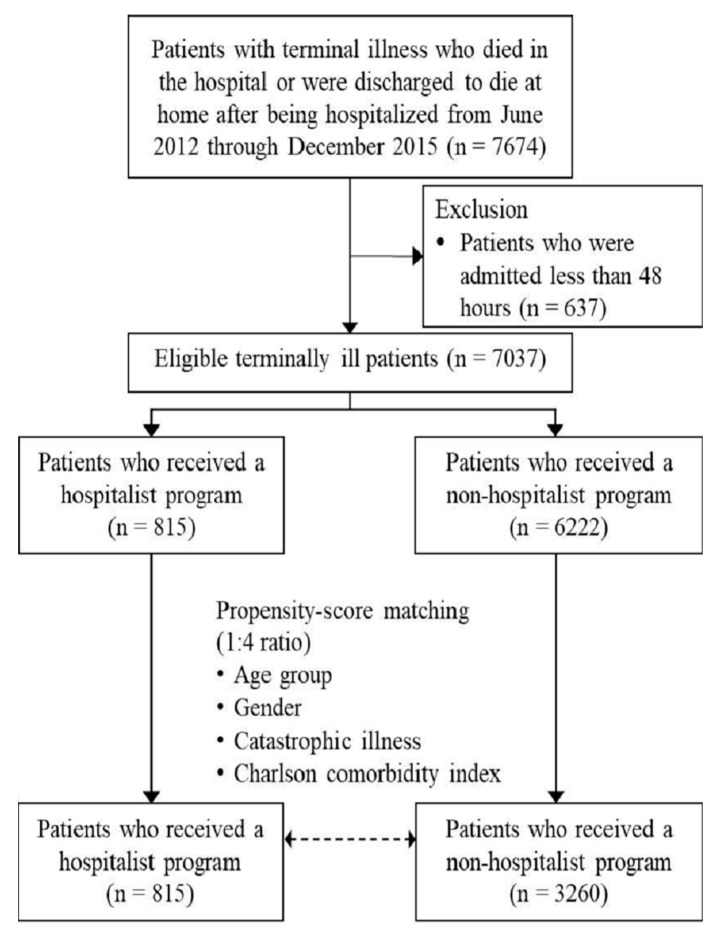
Study diagram.

**Table 1 cancers-15-03976-t001:** Characteristics of patients in this study with and without hospitalist care.

Variables	Before Matching				*p*-Value
	Hospitalist Care (*n* = 815)		Non-Hospitalist Care (*n* = 6222)		
	*n*	(%)	*n*	(%)	
Age (years), median (IQR)	75.6	(63.3–84.6)	65.8	(54.5–78.1)	<0.001
<65	235	(28.8)	2995	(48.1)	<0.001
65–79.9	265	(32.5)	1914	(30.8)	
≥80	315	(38.7)	1313	(21.1)	
Gender					0.174
Female	359	(44.0)	2583	(41.5)	
Male	456	(56.0)	3639	(58.5)	
Catastrophic illness	547	(67.1)	4813	(77.4)	<0.001
CCI score, median (IQR)	2.0	(0–6)	3.0	(0–6)	0.740
0	258	(31.7)	2015	(32.4)	0.008
1–4	296	(36.3)	1987	(31.9)	
5–8	149	(18.3)	1425	(22.9)	
≥9	112	(13.7)	795	(12.8)	
Primary diagnosis					
Neoplasms	243	(29.8)	2694	(43.3)	<0.001
Infectious and parasitic diseases	79	(9.7)	424	(6.8)	0.004
Circulatory system diseases	50	(6.1)	787	(12.7)	<0.001
Respiratory system diseases	244	(29.9)	910	(14.6)	<0.001
Digestive system diseases	80	(9.8)	347	(5.6)	<0.001
Genitourinary system diseases	38	(4.7)	130	(2.1)	<0.001
Injury and poisoning	14	(1.7)	186	(3.0)	0.043
Others	67	(8.3)	744	(11.9)	0.001
Disposition, *n* (%)					0.214
Death	665	(81.6)	5185	(83.3)	
Discharged home to die	150	(18.4)	1037	(16.7)	

**Table 2 cancers-15-03976-t002:** Comparisons of patients with and without hospitalist care after propensity score matching.

Variables	After Matching				*p*-Value
	Hospitalist Care (*n* = 815)		Non-Hospitalist Care (*n* = 3260)		
	*n*	(%)	*n*	(%)	
Age (years), median (IQR)	75.6	(63.3–84.6)	71.4	(62.5–83.8)	0.194
<65	235	(28.8)	939	(28.8)	0.992
65–79.9	265	(32.5)	1067	(32.7)	
≥80	315	(38.7)	1254	(38.5)	
Gender					0.844
Female	359	(44.0)	1449	(44.5)	
Male	456	(56.0)	1811	(55.5)	0.174
Catastrophic illness	547	(67.1)	2297	(70.5)	0.067
CCI score, median (IQR)	2.0	(0–6)	3.1	(0–6)	0.167
0	258	(31.7)	1028	(31.5)	0.983
1–4	296	(36.3)	1198	(36.7)	
5–8	149	(18.3)	602	(18.5)	
≥9	112	(13.7)	432	(13.3)	
Primary diagnosis					
Neoplasms	243	(29.8)	1179	(36.2)	<0.001
Infectious and parasitic diseases	79	(9.7)	242	(7.4)	0.035
Circulatory system diseases	50	(6.1)	485	(14.9)	<0.001
Respiratory system diseases	244	(29.9)	611	(18.7)	<0.001
Digestive system diseases	80	(9.8)	197	(6.0)	<0.001
Genitourinary system diseases	38	(4.7)	87	(2.7)	0.004
Injury and poisoning	14	(1.7)	105	(3.2)	0.020
Others	67	(8.3)	354	(10.9)	0.028
Disposition, *n* (%)					0.839
Death	665	(81.6)	2671	(81.9)	
Discharged home to die	150	(18.4)	589	(18.1)	

**Table 3 cancers-15-03976-t003:** Primary and secondary outcomes between patients with and without hospitalist care.

Outcome Variables	Hospitalist Care (*n* = 815)		Non-Hospitalist Care (*n* = 3260)		*p*-Value
	*n*	(%)	*n*	(%)	
Palliative care consultation	121	(14.9)	164	(5.0)	<0.001
Life-sustaining procedures					
Invasive mechanical ventilation	77	(9.4)	1269	(38.9)	<0.001
Non-invasive ventilation	75	(8.0)	525	(16.1)	<0.001
Tracheostomy	6	(0.7)	151	(4.6)	<0.001
Hemodialysis	89	(10.9)	471	(14.5)	0.009
Surgery	88	(10.8)	1084	(33.3)	<0.001
ICU admission	88	(10.8)	1492	(45.8)	<0.001
Hospitalization resources					
Length of stay (days), median (IQR)	12.0	(6.0–21.0)	19.0	(9.0–36.0)	<0.001
Total medical cost (TWD), median (IQR)	65,590.0	(31,999.0– 142,432.0)	192,745.5	(83,040.5– 427,174.5)	<0.001
Daily medical cost (TWD), median (IQR)	6129.6	(4640.9– 8255.5)	9739.5	(6477.8– 16,943.4)	<0.001

**Table 4 cancers-15-03976-t004:** Univariable and multivariable regression models for primary and secondary outcomes associated with hospitalist care as compared with non-hospitalist care.

	Univariable Model			Multivariable Model *		
	OR	(95% CI)	*p*-Value	OR	(95% CI)	*p*-Value
Palliative care consultation	3.29	(2.56–4.24)	<0.001	3.41	(2.63–4.41)	<0.001
Life-sustaining procedures						
Invasive mechanical ventilation	0.16	(0.13–0.21)	<0.001	0.13	(0.10–0.17)	<0.001
Non-invasive ventilation	0.52	(0.41–0.68)	<0.001	0.51	(0.39–0.66)	<0.001
Tracheostomy	0.15	(0.07–0.35)	<0.001	0.14	(0.06–0.31)	<0.001
Hemodialysis	0.73	(0.57–0.92)	0.009	0.70	(0.55–0.89)	0.004
Surgery	0.24	(0.19–0.31)	<0.001	0.25	(0.19–0.31)	<0.001
ICU admission	0.14	(0.11–0.18)	<0.001	0.11	(0.08–0.14)	<0.001
Hospitalization resource	B	(95% CI)	*p*-value	B	(95% CI)	*p*-value
Length of stay (days)	−0.58	(−0.66–−0.50)	<0.001	−0.54	(−0.62–−0.46)	<0.001
Total medical cost (TWD)	−1.06	(−1.18–−0.95)	<0.001	−0.96	(−1.06–−0.87)	<0.001
Daily medical cost (TWD)	−0.73	(−0.80–−0.66)	<0.001	−0.59	(−0.65–−0.54)	<0.001

CI, confidence interval; ICU, intensive care unit; OR, odds ratio; TWD, Taiwan dollar. * Adjusted for patient’s age, gender, catastrophic illness, primary diagnoses, Charlson comorbidity index score, and disposition status.

## Data Availability

The data presented in this study are available on request from the corresponding author. The data are not publicly available due to reasons of data protection.

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
