# Peer review of "Does Hospitalist Care Enhance Palliative Care and Reduce Aggressive Treatments for Terminally Ill Patients? A Propensity Score-Matched Study"

_cancers, 2023, doi:10.3390/cancers15153976_

Round 1

Reviewer 1 Report

Title: the case was defined as a group of patients cared by a hospitalist and it was not outcome. Thus, the could its design be a case-cohort study rather than case control ?

Abstract: in the results, it may be better to describe “proportion” for “probability.”

Introduction: in the last sentence, this study was based on observational design and thus it would be appropriate to use “association” rather than “improve” or “reduce”.

Methods: line 154: typo? principle; maybe principal?

Methods and Figure 1: in the uppermost box, patients were described as “terminal illness”. But did the cohort include patients with potentially-curable acute illness? How about those with young and CCI of zero? Any data for stage and performance stays in those with neoplasms?

Author Response

.

Reviewer 2 Report

Overall, this is an innovative and well-conducted study, and the target topic is really important. The writing is clear, the methodology used is adapted to the questions asked, and the results align with the authors' conclusions.  Thus, I have only minor remarks concerning this article which are listed below.

- Change and check the keywords with the MeSH term via this link: https://www.ncbi.nlm.nih.gov/mesh.

- The conclusion of abstract is the repetition of what has been said in the text! And need a revision based on the results.

- The discussion is disorganized and confusing. It is better to summarize your findings first and then discuss them separately and focus on findings in East Asia countries.

- Minor editing of English language required.

Reviewer 3 Report

This is an interesting retrospective, single Institution study on palliative care utilization and rate of aggressive treatment at the end of life in terminally ill patients receiving hospitalist care.

815 patients who received care from hospitalists between June 1, 2012 and December 31, 2015 were included in this study and 6222 patients who did not receive hospitalist care were utilized as control group

Authors reported in the method section that the decision to admit patients to hospitalist wards is made by emergency physicians, however, the criteria for admission on Hospitalist ward are not clearly described.

The characteristics of patients admitted in hospitalist ward reported in table 1 (before propensity score matching) clearly show different in severity, age and pathology respect to control group.

Although the results of the study, after selection of control group according to propensity score maching, show a significant higher proportion of palliative care consultation during hospitalist care (14.9 vs 5%), in my opinion, the rate of palliative care consultation in a population of terminally ill patients is very low in both group.

I suggest to better describe palliative care intervention. Patients received only one palliative care visit or more? 

Palliative care consultation involved patients and families and were focused on end of life treatment patients’ preferences or to collect Advance Care Planning?

The results show also a significant reduction of aggressive life-sustaining procedures in hospitalist care assisted patient.

According to the authors, hospitalist-led multidisciplinary team approach for terminally ill patients is the key to enhance palliative care and reduce the utilization of life-sustaining procedures and hospitalization resources during the EOL stage.

However, quality of care at the end of life should not be based only on physician attitude but also to an involvement of patients and families in end of life treatment decision.

This crucial aspect should be reported in the discussion section

Author Response

Please see the attachment including our response to your comment. Thank you.
